# Effectiveness in Block by Dexmedetomidine of Hyperpolarization-Activated Cation Current, Independent of Its Agonistic Effect on α_2_-Adrenergic Receptors

**DOI:** 10.3390/ijms21239110

**Published:** 2020-11-30

**Authors:** Te-Ling Lu, Te-Jung Lu, Sheng-Nan Wu

**Affiliations:** 1School of Pharmacy, China Medical University, Taichung City 406040, Taiwan; lutl@mail.cmu.edu.tw; 2Department of Medical Laboratory Science and Biotechnology, Chung Hwa University of Medical Technology, Tainan City 71703, Taiwan; lutejung@yahoo.com.tw; 3Department of Medical Research, China Medical University Hospital, China Medical University, Taichung City 40402, Taiwan; 4Institute of Basic Medical Sciences, National Cheng Kung University Medical College, Tainan City 70101, Taiwan; 5Department of Physiology, National Cheng Kung University Medical College, No. 1, University Road, Tainan City 70101, Taiwan

**Keywords:** dexmedetomidine, endocrine cells, hyperpolarization-activated cation current, voltage hysteresis, action potential

## Abstract

Dexmedetomidine (DEX), a highly selective agonist of α_2_-adrenergic receptors, has been tailored for sedation without risk of respiratory depression. Our hypothesis is that DEX produces any direct perturbations on ionic currents (e.g., hyperpolarization-activated cation current, *I*_h_). In this study, addition of DEX to pituitary GH_3_ cells caused a time- and concentration-dependent reduction in the amplitude of *I*_h_ with an IC_50_ value of 1.21 μM and a *K*_D_ value of 1.97 μM. A hyperpolarizing shift in the activation curve of *I*_h_ by 10 mV was observed in the presence of DEX. The voltage-dependent hysteresis of *I*_h_ elicited by long-lasting triangular ramp pulse was also dose-dependently reduced during its presence. In continued presence of DEX (1 μM), further addition of OXAL (10 μM) or replacement with high K^+^ could reverse DEX-mediated inhibition of *I*_h_, while subsequent addition of yohimbine (10 μM) did not attenuate the inhibitory effect on *I*_h_ amplitude. The addition of 3 μM DEX mildly suppressed the amplitude of *erg*-mediated K^+^ current. Under current-clamp potential recordings, the exposure to DEX could diminish the firing frequency of spontaneous action potentials. In pheochromocytoma PC12 cells, DEX was effective at suppressing *I*_h_ together with a slowing in activation time course of the current. Taken together, findings from this study strongly suggest that during cell exposure to DEX used at clinically relevant concentrations, the DEX-mediated block of *I*_h_ appears to be direct and would particularly be one of the ionic mechanisms underlying reduced membrane excitability in the in vivo endocrine or neuroendocrine cells.

## 1. Introduction

Dexmedetomidine (DEX, (S)-4-[1-(2,3-dimethylphenyl)ethyl]-1H-imidazole hydrochloride, Precedex^®^), a lipophilic imidiazole derivative, is regarded as a potent and selective agonist of the α_2_-adrenergic receptors [1]. It has been revealed to exert a variety of actions on the human brain such as sedation, anesthetic-sparing effects, and analgesia [1,2]. However, there is a growing evidence to highlight the notion that the direct interactions with membrane ion channels may be an unidentified but important mechanism underlying DEX-induced actions in central neurons [3,4,5]. A recent report also showed that guanabenz, another α_2_-adrenoceptor agonist, could directly suppress hyperpolarization-activated cation current in mesencephalic trigeminal nucleus neurons [5]. DEX at high concentrations was previously reported to suppress compound action potentials in frog sciatic nerve, and this action is thought to be independent on its activation of the α_2_-adrenergic receptor [4]. Alternatively, DEX was shown to increase the expression of phosphorylated ERK1 and 2 in the hippocampus, a key element in signal transduction, and this action was also independent from its binding to the α_2_-adrenerginic receptor [6].

However, an earlier report showed that the ability of norepinephrine to suppress prolactin secretion at the pituitary gland is mediated by the binding to adrenergic receptors [7]. Another previous work reported the ability of DEX to suppress hyperpolarization-activate cation current (*I*_h_) and this effect tended to be reversed by further addition of α_2_-adrenergic receptor blocker [8]. It has been also recently demonstrated to exert anticonvulsant and neuroprotective effects in kainate-treated rats [9]. DEX-mediated inhibition of voltage-gated Na^+^ current seen in dorsal root ganglion neurons was thought to be mediated through its binding to α_2_-adrenergic receptors [10]. Therefore, the perturbating effects of DEX on different types of ionic currents have been thus far incompletely understood.

Hyperpolarization-activated cation current (*I*_h_) has been recognized as a key determinant of repetitive electrical activity in heart cells and in a variety of neurons, and neuroendocrine or endocrine cells [11,12,13,14,15,16,17]. This current is a mixed inward Na^+^/K^+^ current (i.e., unusual ion selectivity in that it conducts both Na^+^ and K^+^ ions), which is sensitive to block by CsCl or ivabradine [14,17,18]; and activation of this current at resting membrane potentials can result in a net inward current carried largely by Na^+^, which will depolarize the membrane potential to the threshold required for the elicitation of action potential (AP) [17,18]. It was assumed to be carried by channels of the hyperpolarization-activated cyclic nucleotide-gated (HCN) gene family, which belongs to the superfamily of voltage-gated K^+^ channels and cyclic nucleotide-gated channels. To date, how DEX can interact with HCN channels to modify the amplitude and/or gating of *I*_h_ still remains unexplored.

Therefore, the hypothesis in the present study is that DEX has direct perturbations on different types of ionic currents including *I*_h_ and *I*_K(erg)_ inherently existing in pituitary GH_3_ cells and further to determine the mechanisms of this drug interacting with this current. Of importance, findings from our experimental data highlight the notion that DEX can directly produce a depressant action on *I*_h_ in a concentration-, time-, and state-dependent manner in these cells (i.e., GH_3_ and PC12 cells). The major depressant action of this drug on *I*_h_ reported herein is thought to be direct and predominantly through its interaction with the open state of the HCN channel.

## 2. Materials and Methods

### 2.1. Chemicals and Solutions

Dexmedetomidine (DEX, dexmedetomidina, dexmedetomidinum, (S)-4-[1-(2,3-dimethylphenyl)ethyl]-1H-imidazole hydrochloride, MPV-1440, C13H16N2, Precedex^®^) was obtained from Abbott Laboratories (Abbott Park, IL), oxaliplatin (OXAL) was obtained from Sanofi-Aventis (New York, NY, USA), ivabradine was obtained from Sigma-Aldrich (St. Louis, MO, USA), and yohimbine was acquired from Tocris (Bristol, UK). Azimilide was a gift from Procter and Gamble Pharmaceuticals (Cincinnati, OH, USA), while chlorotoxin was from Professor Dr. Woei-Jer Chuang (Department of Biochemistry, National Cheng Kung University Medical College, Tainan City, Taiwan). Ham’s F-12 and RPMI 1640 medium, horse serum, fetal calf serum, L-glutamine, and trypsin/EDTA were obtained from Invitrogen (Carlsbad, CA, USA), while other chemicals that include CsCl, HEPES, and aspartic acid were at analytical grades.

The HEPES-buffered normal Tyrode’s solution contained (in mM): NaCl 136.5, KCl 5.4, CaCl_2_ 1.8, MgCl_2_ 0.53, glucose 5.5, and HEPES-NaOH buffer 5.5 (pH 7.4). To record membrane potential or *I*_h_ and to adequately avoid contamination of Cl- currents, the pipette was filled with solution (in mM): K-aspartate 130, KCl 20, KH_2_PO_4_ 1, MgCl_2_ 1, EGTA 0.1, Na_2_ATP 3, Na_2_GTP 0.1, and HEPES-KOH buffer 5 (pH 7.2) [19], which was filtered on the day with 0.22 μm filter (Millipore). The high-K^+^, Ca^2+^-free solution was used to measure *erg*-mediated K^+^ current (*I*_K(erg)_) containing (in mM): KCl 145, MgCl_2_ 0.53, and HEPES-KOH buffer 5 (pH 7.4).

### 2.2. Cell Preparations

GH_3_, a rat pituitary cell line derived from a pituitary tumor (BCRC-60015), and rat pheochromocytoma PC12 cells (BCRC-60048) were obtained from the Bioresources Collection and Research Center (Hsinchu, Taiwan). Briefly, we cultured GH_3_ cells in Ham’s F-12 medium supplemented with 2.5% fetal calf serum (*v*/*v*), 15% heat-inactivated horse serum (*v*/*v*), and 2 mM L-glutamine. To promote optimum differentiation, GH_3_ cells were transferred to a serum-free, Ca^2+^-free medium (Wu et al., 2017). PC12 cells were cultured in RPMI 1640 medium supplemented with 5% fetal bovine serum (*v*/*v*) and 10% horse serum (*v*/*v*). The cells were grown in a humidified 5% CO_2_/95% air incubator and they commonly underwent passage every 2 weeks. Cell viability was commonly evaluated by the trypan blue dye-exclusion test. Under the experimental conditions studied, cells remained 80–90% viable for at least 2 weeks. Patch-clamp experiments were performed 5 or 6 days after cells were cultured (60–80% confluence).

### 2.3. Electrophysiological Measurements

On the day of the experiments, cells (i.e., GH_3_ or PC12 cells) were harvested and a few drops of cell suspension was immediately transferred to a home-made recording chamber affixed to the stage of a DM-IL inverted microscope (Leica, Wetzlar, Germany). Cells visualized under inverted microscope were immersed at room temperature in normal Tyrode’s solution, the composition of which is detailed above. We performed the patch-clamp recordings under whole-cell configuration with either an RK-400 (Biologic, Echirolles, France) amplifier, or an Axopatch-200B or Axoclamp-2B (Molecular Devices, Sunnyvale, CA, USA) amplifier [20]. The recording pipette electrodes with tip resistances of 3–5 MΩ were prepared from Kimax-51 glass capillaries (#34500 (outer diameter: 1.5–1.8 mm); Kimble Products, Vineland, NJ, USA), and we fabricated the capillary tubes by using either a PP-83 vertical puller (Narishige, Tokyo, Japan) or a P-97 horizontal puller (Sutter, Novato, CA, USA). An electrode holder filled with a silver chloride-coated silver wire connected the patch electrode to the amplifier. During the measurements, the electrode used was mounted on and delicately controlled by a WR-98 hydraulic micromanipulator (Narishige).

### 2.4. Data Recordings

The signals, which include voltage and current tracings, were stored online through analog-to-digital conversion at 10 kHz in an Acer SPIN-5 touchscreen laptop computer (SP51–52N-55WE; Taipei City, Taiwan) [20]. The laptop computer that was controlled by pCLAMP 10.7 software (Molecular Devices) was put on top of adjustable Cookskin stand (Ningbo, Zhejiang, China) for efficient heat dissipation of the computer and program controlling during the recordings. Current signals were low-pass filtered at 2 kHz with a FL-4 four-pole Bessel filter (Dagan, Minneapolis, MN, USA) to minimize background noise. By use of digital-to-analog conversion, we devised the pCLAMP-generated voltage-clamp profiles that include various rectangular or ramp waveforms, to establish the current–voltage (*I*–*V*) relationships, steady-state activation curve, and voltage hysteresis of ionic currents (e.g., *I*_h_). When high-frequency stimuli were needed, we equipped an Astro-Med Grass S85X dual output pulse stimulator (Grass Technologies, West Earwick, RI, USA). The digitized results in this study were sent to a remote cloud site for further analyses, data sharing, and archiving.

### 2.5. Data Analyses

To establish concentration-dependent inhibition of DEX on the amplitude of *I*_h_ elicited in response to pulsed hyperpolarization, cells were immersed in Ca^2+^-free Tyrode’s solution, and each cell examined was voltage-clamped at −40 mV and the 2-s hyperpolarizing pulse from −40 to −110 mV was thereafter delivered at a rate of 0.05 Hz to evoke *I*_h_. Current amplitudes measured at the end of each hyperpolarizing step were taken and compared in the control and during the exposure to varying concentrations (0.3–10 μM) of DEX. We determined the concentration needed to suppress 50% of current amplitude with the goodness of fit by using the multiparameter logistic equation [21,22]:$$\mathrm{Relative}\;\mathrm{amplitude}=\frac{\left(1-a\right)\times {\left[C\right]}^{-n_{H}}}{{\left[C\right]}^{-n_{H}}+{\mathrm{IC}}_{50}^{-n_{H}}}+a,$$where [C] represents the DEX concentration; IC_50_ and n_H_ are the concentration required for a 50% inhibition and the Hill coefficient, respectively; and maximal inhibition (i.e., 1-a) was also estimated in this equation.

Since the depressant effect of DEX on *I*_h_ measured from GH_3_ cells is thought to arise from a state-dependent blocker that preferentially binds to the open state of the channel, a minimal first-order scheme was thereafter derived as the following:
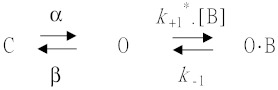
where α and β represent the voltage-gated rate constants for the opening and closing of HCN channels elicited by step hyperpolarization, respectively; [B] is the DEX concentration; *k*_+1_* or *k*_−1_ is rate constant for blocking or unblocking produced by the presence of DEX; and C, O, or O·B in each term indicates the closed, open, or open-blocked state, respectively.

The blocking (i.e., on) or unblocking (i.e., off) rate constant, *k*_+1_* or *k*_−1_, was assessed from activation time constant (τ) of hyperpolarization-elicited relative block (i.e., (*I*_control_−*I*_DEX_)/*I*_control_) of *I*_h_ collected during the exposure to different concentrations of DEX. These rate constants from the above-described reaction scheme were allowed to be optimized using the relation:$$\frac{1}{\tau }=k_{+1}{}^{*}\times [B]+k_{-1},$$where *k*_+1_* or *k*_−1_ was respectively taken from the slope or from the *y*-axis intercept at [B] = 0 of the linear regression that interpolates the reciprocal time constants (i.e., 1/τ) versus the DEX concentration, and [B] is the DEX concentration applied.

The steady-state activation curve of *I*_h_ with or without the application of DEX was determined. The sigmoidal curves of the current were least-squares fitted by the Boltzmann function:$$\frac{I}{I_{max}}=\frac{1}{1+exp\left[\frac{\langle V-V_{1/2}\rangle qF}{RT}\right]},$$where *I*_max_ = the maximal *I*_h_ amplitude, *V*_1/2_ = the voltage at which there is half-maximal activation of the current, *q* = the apparent gating charge, namely the charge across the membrane electric field between closed and open conformations, *R* = the universal gas constant, *F* = Faraday’s constant, and *T* = the absolute temperature.

### 2.6. Statistical Analyses

The pertinent data with respect to linear or nonlinear curve-fitting (e.g., exponential or sigmoidal curve) was processed, and the curves were then constructed with least-squares minimization procedure by using Microsoft Solver function embedded in Excel 2019 (Microsoft) or 64-bit OriginPro 2016 program (OriginLab). All numerical data are presented as the mean ± standard error of the mean (SEM) with sample sizes (*n*) indicating the number of cells. The paired or unpaired Student’s *t*-test and one-way analysis of variance (ANOVA) were utilized for statistical analyses; however, assuming that the statistical difference among different groups was necessarily evaluated, post-hoc Duncan’s multiple comparisons were further performed. Statistical analyses were performed using the SPSS 20 statistical software package (IBM Corp., Armonk, NY, USA). Statistical significance was determined at a *p*-value of <0.05.

## 3. Results

### 3.1. Effect of DEX on Hyperpolarization-Activated Cation Current (I_h_) Recorded from GH_3_ Cells

In the initial stage of experiments, we tested whether the presence of DEX could cause any possible modifications on the amplitude or gating of *I*_h_ elicited by membrane hyperpolarization. We bathed GH_3_ cells in Ca^2+^-free Tyrode’s solution and then filled the recording pipette by using high K^+^-containing solution as detailed above. As the whole-cell model was achieved, we voltage-clamped the examined cell at −40 mV and subsequently applied a hyperpolarizing command potential to −100 mV with a duration of 2 s. In good agreement with those described before in different cell types that include heart and pituitary cells [11,13,14,23,24], the *I*_h_ during long-lasting hyperpolarization was observed to display an inward current with the slowly activating time course and inwardly rectifying property. Of considerable interest, as cells were exposed to different DEX concentrations, the amplitude of *I*_h_ from −40 to −100 mV with a duration of 2 s progressively became decreased (Figure 1A). For example, the addition of 1 μM DEX decreased *I*_h_ amplitude from 253.3 ± 12.5 to 138.7 ± 9.8 pA (*n* = 11, *p* < 0.05). After washout of the agent, current amplitude returned to 191.7 ± 10.2 pA (*n* = 9, *p* < 0.05). Moreover, the activation time constant (τ) of *I*_h_ in the presence of DEX became increased in a concentration-dependent manner (Figure 1B). For example, addition of 1 μM significantly raised the τ value of *I*_h_ activation from 683 ± 16 to 1407 ± 29 msec (*n* = 9, *p* < 0.05).

It was noticed that the inhibitory effects of DEX on hyperpolarization-elicited *I*_h_ in GH_3_ cells do not emerge instantaneously (i.e., they vary in time), but they occur in a time- and concentration-dependent fashion. In this regard, to provide quantitative evaluation of DEX-induced block of *I*_h_, we continued to analyze the time constants for the relative block of *I*_h_ (i.e., (*I*_control_−*I*_DEX_)/*I*_control_) observed in these cells. The time courses for the relative block in the presence of different DEX concentrations were fitted by use of a single-exponential function. The concentration dependence of the relative block for *I*_h_ in response to membrane hyperpolarization is illustrated in Figure 1C. It is evident from these results that the exposure to DEX led to a concentration-dependent increase in the rate (1/τ) of the relative block. There was a linear relationship between the 1/τ value and the DEX concentration with a correlation coefficient of 0.95 (Figure 1D). The blocking and unblocking rate constants directly derived from minimum kinetic scheme as described in the Materials and Methods were consequently estimated to be 2.419 s^−1^μM^−1^ and 4.77 s^−1^, respectively; thereafter, the value of dissociation constant, namely *K*_D_ = *k*_−1_/*k*_+1_*, was estimated to yield 1.98 μM. Moreover, the concentration-dependent effect of DEX on *I*_h_ amplitude measured at the end of hyperpolarizing pulse was constructed and is illustrated in the present study (Figure 1D). The effective IC_50_ value required for DEX-mediated inhibition of *I*_h_ amplitude in GH_3_ was computed to be 1.21 μM (Figure 1E), a value that is nearly similar to the *K*_D_ value determined by the binding scheme. The averaged *I–V* relationships of *I*_h_ measured at the beginning or end of hyperpolarizing pulses in the control and during the exposure to 1 μM DEX were depicted in Figure 2. On the basis of these *I*–*V* curves, the macroscopic *I*_h_ conductance measured at the end of voltage steps ranging between −100 and −130 mV was conceivably decreased to 2.73 ± 0.04 nS from a control value of 4.38 ± 0.07 nS (*n* = 9, *p* < 0.05)

### 3.2. Effect of DEX on the Steady-State Activation Curve of i_h_

To characterize the inhibitory effect of DEX on *I*_h_, we further studied whether this drug might affect any changes in the steady-state activation curve of *I*_h_ in GH_3_ cells. Figure 2D illustrates the activation curves of *I*_h_ obtained in the absence and presence of DEX (1 μM) taken from GH_3_ cells. The Boltzmann distribution detailed in the Materials and Methods was implemented to fit the data points properly. In control, V_1/2_ = −94.7 ± 3.4 mV and q = 2.45 ± 0.09 *e* (*n* = 7), and in the presence of 1 μM DEX, V_1/2_ = −105.1 ± 3.6 mV and 2.42 ± 0.09 *e* (*n* = 7). The results reflected that during the exposure to 1 μM DEX, the activation curve of *I*_h_ was shifted along the voltage axis toward more hyperpolarized potential by roughly 10 mV, albeit without significant change in the gating charge of the curve.

### 3.3. Effect of DEX on the Voltage-Dependent Hysteresis Elicited in Response to Long-Lasting Triangular Ramp Pulse

The voltage-dependent hysteresis of ionic currents (e.g., *I*_h_) has been previously characterized to exert a significant impact on electrical behaviors such as AP firing [25,26,27]. In this regard, we continued to explore whether there is possible voltage-dependent hysteresis existing in *I*_h_ recorded from GH_3_ cells. In this set of current recordings, we applied a slow triangular ramp pulse with duration (i.e., 0.29 V/s) to the examined cell, as whole-cell configuration was achieved. It is noticeable from Figure 3A that the trajectories of *I*_h_ responding to the upsloping (i.e., depolarizing from −140 to −40 mV) and downsloping (hyperpolarizing from −40 to −140 mV) ramp command as a function of time were apparently distinguishable between them. In other words, the current amplitude elicited by the upsloping limb of triangular voltage ramp was higher than that during the downsloping limb, hence enabling us to indicate that there is a voltage hysteresis for this current as demonstrated in Figure 3B, namely the relationship of *I*_h_ versus membrane potential. As the ramp speed became reduced, the hysteresis degree for *I*_h_ was progressively raised. Of notice, as the examined cell was exposed to DEX (1 μM), *I*_h_ amplitude evoked in the upsloping limb of long-lasting triangular ramp was detected to decrease to a greater extent than that measured from the downsloping limb. For example, in controls, *I*_h_ at the level of −100 mV elicited upon the upsloping and downsloping limbs of triangular ramp pulse were 118 ± 18 and 68 ± 9 pA (*n* = 9), respectively, the values of which were observed to differ significantly between them (*p* < 0.05). Moreover, in the presence of 1 μM DEX, the amplitudes of forward and backward *I*_h_ taken at the same level of membrane potential were significantly reduced to 77 ± 18 and 59 ± 12 pA (*n* = 9, *p* < 0.05), respectively.

We continued to quantify the degree of voltage-dependent hysteresis on the basis of the observed difference in area under the curves in the forward (upsloping) and reverse (downsloping) direction as denoted by the arrows in Figure 3B. It was observed that for *I*_h_ in GH_3_ cells, the strength of voltage hysteresis highly increased with slower ramp speed, and that the presence of DEX resulted in a reduction in the magnitude of such hysteresis. Figure 3C illustrates summary of the data showing effects of DEX at the different concentrations on the area under the curve between forward and backward current traces. For example, apart from its suppression of *I*_h_ magnitude, addition of DEX (1 μM) significantly decreased the area by about 50% elicited in response to such slow triangular voltage ramp.

### 3.4. Effects of DEX, DEX Plus Oxaliplatin (OXAL), DEX Plus High K, and DEX Plus Yohimbine on the Amplitude of I_h_ in GH_3_ Cells

We next explored whether the suppressive effect of DEX on *I*_h_ could be modified by subsequent addition of OXAL, high-K^+^ solution, or yohimbine. OXAL was recently reported to elevate *I*_h_ amplitude [28], challenging cells with high K^+^ solution has been shown to increase *I*_h_ amplitude [12,29,30], and yohimbine was reported to be an antagonist of α_2_-adrenergic receptors [31]. As revealed in Figure 4A,B, subsequent application of OXAL, still in continued presence of 1 μM DEX, was efficacious at attenuating DEX-mediated suppression of *I*_h_ measured at the end of the hyperpolarizing voltage pulse. As cells were continually exposed to 1 μM DEX, a further rise of extracellular K^+^ concentration to 20 mM also counteracted *I*_h_ suppressed by DEX (Figure 5B). However, unlike the experimental results made in previous observations [8], subsequent addition of 10 μM yohimbine, an antagonist of α_2_-adrenergi receptor, did not result in any alterations on DEX-induced inhibition of *I*_h_ in GH_3_ cells.

### 3.5. Effect of DEX on Erg-Mediated K^+^ Current (I_K(erg)_) in GH_3_ Cells

In a separate set of experiments, we continued to test whether the presence DEX can modulate different type of K^+^ currents, (i.e., *I*_K(erg)_), identified in these cells. The experiments were conducted in cells bathed in high-K^+^, Ca^2+^-free solution and the recording electrode used was filled up with the K^+^-containing solution. As illustrated in Figure 5, DEX at a concentration of 3 μM was noted to mildly suppress the amplitude of deactivating *I*_K(erg)_, as revealed by a significant reduction of current amplitude from 752 ± 45 to 612 ± 33 pA (*n* = 8, *p* < 0.05). Moreover, still in the presence of DEX, further addition of azimilide (10 μM), an inhibitor of *I*_K(erg)_, was capable of depressing *I*_K(erg)_ significantly to 225 ± 24 pA (*n* = 8, *p* < 0.05). Therefore, as compared with its suppressive action on *I*_h_, the *I*_K(erg)_ in GH_3_ cells appear to be less vulnerable to suppression by DEX.

### 3.6. Effect of DEX on Spontaneous Action Potentials (APs) Recorded from GH_3_ Cells

We also determined how the exposure to DEX could perturb spontaneous APs recorded under the current-clamp condition. This set of experiments was conducted on cells immersed in normal Tyrode’s solution containing 1.8 mM CaCl_2_. As revealed in Figure 6, when the whole-cell voltage-clamp mode was securely established and then rapidly switched to the current-clamp potential recordings, the application of DEX caused a concentration-dependent effect on the suppression of the frequency in spontaneous APs in GH_3_ cells, together with membrane depolarization. In continued presence of DEX (1 μM), subsequent application of ivabradine (1 μM), an effective inhibitor of *I*_h_ [24], decreased firing frequency further; however, further addition of yohimbine (10 μM) exerted a minimal effect on it. It is thus possible from the current experiment that DEX-mediated suppression of firing frequency in these cells is primarily explained by its inhibitory actions on the amplitude and gating of *I*_h_ described above, yet not solely by the agonistic effect on α_2_-adrenergic receptors.

### 3.7. Effect of DEX and DEX Plus Ivabradine on I_h_ Recorded from Pheochromocytoma PC12 Cells

A current report showed that DEX could protect lidocaine-induced neurotoxicity in PC12 cells [32]. Therefore, in a final set of experiments, we also evaluated the effects of DEX on *I*_h_ in another type of endocrine cells (i.e., pheochromocytoma PC12 cells). PC12 cells were bathed in Ca^2+^-free Tyrode’s solution and the recording pipette was filled with the K^+^-containing solution. Indistinguishable from the biophysical properties of *I*_h_ measurable in GH_3_ cells described above, the *I*_h_ was evoked as the examined PC12 cell was hyperpolarized from −40 to −100 mV with a duration of 2 s. As illustrated in Figure 7, DEX at a concentration of 1 μM effectively suppressed *I*_h_ amplitude measured during long-step pulsed hyperpolarization to −100 mV. The *I*–*V* relationships of *I*_h_ taken with or with DEX (1 μM) addition were constructed and are then illustrated in Figure 8. The presence of 1 μM DEX significantly decreased whole-cell *I*_h_ conductance obtained at the voltages between −100 and −120 mV from 3.45 ± 0.12 to 1.85 ± 008 nS (*n* = 8, *p* < 0.05). Moreover, subsequent application of 1 μM ivabradine, but still in the presence of DEX, was observed to depress *I*_h_ amplitude further. Ivabradine was previously reported to suppress *I*_h_ [18,24,33]. Therefore, in accordance with DEX effects on *I*_h_ described above in GH_3_ cells, its presence is effective at inhibiting *I*_h_ observed in PC12 cells. DEX and ivabradine are capable of exerting synergistic effects on the inhibition of *I*_h_ magnitude.

## 4. Discussion

Findings from the present study are as follows: (**a**) in pituitary GH_3_ cells, DEX produced a depressant action on *I*_h_ in a concentration- and state-dependent fashion with an IC_50_ or *K*_D_ values of 1.21 or 1.97 μM, respectively; (**b**) the presence of DEX shifted the steady-state activation curve of *I*_h_ along the voltage axis to more hyperpolarized potential; (**c**) this agent was capable of diminishing the voltage-dependent hysteresis of *I*_h_ during long-lasting triangular (i.e., upsloping and downsloping) ramp pulse, (**d**) it mildly suppressed deactivating *I*_K(erg)_, (**e**) it diminished the firing frequency of spontaneous APs detected under current-clamp conditions, and (**f**) in pheochromocytoma PC12 cells, this drug also suppressed *I*_h_ effectively. The results led us to reflect that, except for its ability to bind to α_2_-adrenergic receptors, the DEX molecule may profoundly and directly act on the activation process of the channel to modify the magnitude and kinetics of *I*_h_ in response to long-lasting membrane hyperpolarization. These are important observations, because DEX-induced block of *I*_h_ shown herein can be a potential ionic mechanism through which it may be efficacious in depressing intrinsic membrane excitability of neurons, or neuroendocrine or endocrine cells in vivo.

It is interesting to note that the effect of DEX on *I*_h_ time course is continuously changing over time. Cell exposure to DEX tended to slow the time course of *I*_h_ activation during long-lasting membrane hyperpolarization, suggesting that such molecule has the propensity to reach the blocking site only when the channel resides in the open state (i.e., the activating state during long-step hyperpolarization). As an alternative, the DEX molecule might have a higher affinity toward the open HCN channels than toward the close (or resting) channels in GH_3_ cells. Therefore, in addition to effective inhibition of *I*_h_ amplitude, the DEX molecule may interact largely on the activation process to influence the modifications on both the magnitude and kinetics of the current. Indeed, based on quantitative estimate from the first-order reaction scheme described herein, the *K*_D_ values required for DEX-induced relative block of *I*_h_ was estimated to 1.97 μM.

In this study, the presence of DEX also exerted an inhibitory action on *I*_h_ in a voltage-dependent fashion. The steady-state activation curve of *I*_h_ during its exposure shifted toward a hyperpolarized potential, with no adequate change in the gating charge of the curve. Therefore, the inhibitory action of this drug would depend on the pre-existing level of resting potential, the pattern of the firing pattern, the DEX concentration achieved, or that occurring in any combinations. However, to what extent DEX or other structurally similar compounds can alter the movement within the S4 helix in voltage-sensing domain of HCN channels as described currently [34] remains to be further explored.

In this study, concentration-dependent inhibition of *I*_h_ with an IC_50_ value of 1.21 μM was observed in the presence of DEX. As cells were exposed to DEX, the amplitude of *I*_h_ was decreased in combination with a substantial increase in the τ value of *I*_h_ activation in response to membrane hyperpolarization. Furthermore, according to the heuristic reaction scheme, the *K*_D_ value was estimated to be 1.97 μM. This value is unlikely to differ from the IC_50_ value (1.21 μM) taken from the concentration-dependent effect of DEX on the suppression of *I*_h_. Since the concentration immediately after a bolus injection with DEX can achieve a range from 0.1 to 1 μM [35], the DEX effects reported herein are thus likely to be therapeutically relevant.

Voltage-dependent hysteresis of *I*_h_ is thought to play substantial role in influencing electrical behavior of electrically excitable cells such as GH_3_ and PC12 cells. In accordance with previous observations [25,26,27], the *I*_h_ natively existing in GH_3_ cells was noticed to undergo either a hysteresis in its voltage dependence, or mode shift in the situation where the voltage sensitivity of gating charge movements relies on the previous state [25,26]. In this study, we also continued to examine the possible perturbation of DEX on such a non-equilibrium property of *I*_h_ in GH_3_ cells. Our results clearly unveiled that the presence of this agent was capable of diminishing such hysteresis linked to the voltage-dependent elicitation of *I*_h_. However, further application of yohimbine, still in the presence of DEX, did not attenuate DEX-mediated reduction in the area of voltage-dependent hysteresis.

Distinguishable from previous observations [8], subsequent addition of yohimbine, an antagonist of α_2_-adrenergic receptors [31], was not observed to attenuate DEX-mediated inhibition of *I*_h_ observed in GH_3_ cells. However, in the continued presence of DEX, we did find out that either further presence of OXAL or replacement with high K^+^ solution (20 mM) has capacity to remain effective at attenuating the block of *I*_h_ induced by DEX. OXAL was recently reported to be effective at stimulating *I*_h_ [28]. The inhibition of *I*_h_ produced by DEX in GH_3_ and PC12 cells appears to be independent of its interaction with α_2_-adrenergic receptors, though pituitary cells were previously reported to intrinsically express those receptors [7].

There are four mammalian subtypes (HCN1, HCN2, HCN3, and HCN4), which have been cloned to date [15,17,26]. It is possible that HCN2, HCN3, or mixed HCN2+HCN3 channels are intrinsically expressed in GH_3_ cells or other types of endocrine or neuroendocrine cells [16,17]. Due to the importance of *I*_h_ (i.e., HCNx-encoded currents) in contributing to the occurrence of excitability and automaticity in electrically excitable cells [13,16,17,18,23], findings from this study could provide novel insights into electrophysiological and pharmacological properties of DEX and other structurally related compounds. Meanwhile, under our current-clamp recordings, addition of DEX was found to reduce the firing frequency of APs recorded from GH_3_ cells. The reduction of AP firing caused by DEX could be principally explained by its clear modifications in the amplitude and gating of *I*_h_. However, it is important to note that the inhibitory effects of DEX on *I*_h_ seen in GH_3_ or PC12 cells tend to be not isoform-specific. Whether DEX-induced bradycardia or different cardioprotective effect [2,36,37] is pertinent to its inhibitory effect on the amplitude and/or kinetic gating of *I*_h_ intrinsically in heart cells also warrant further investigations.

In light of the present findings, the perturbations by DEX of membrane ionic currents such as *I*_h_ and *I*_K(erg)_ are important and thus definitely lend credence to the notion that such actions might be in part connected with its neurological or adverse effects, though the detailed ionic mechanism of its actions remains further clarified. Awareness thus needs to be strengthened in attributing its aberrant use specifically to the selective agonistic effect on α_2_-adrenergic receptors [1,7,8,9,32,38,39].

## Figures and Tables

**Figure 1 ijms-21-09110-f001:**
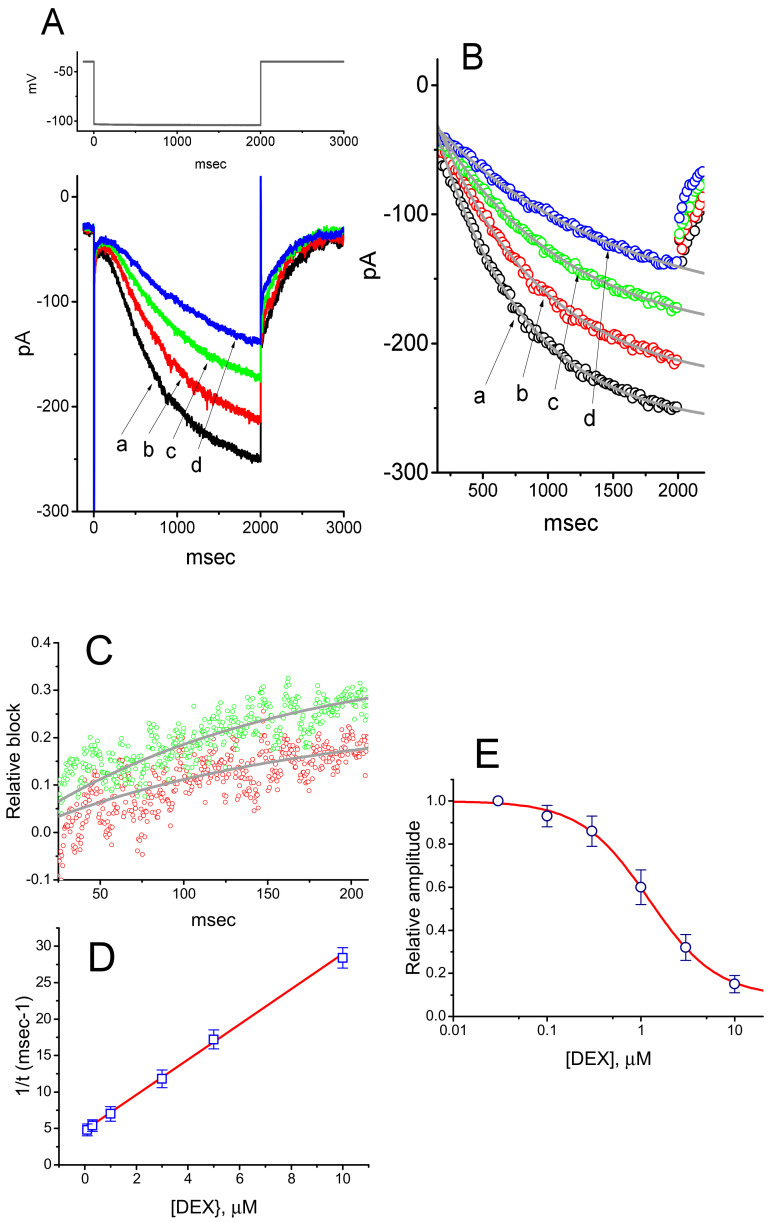
Effect of DEX on hyperpolarization-activated cation current (*I*_h_) recorded from GH_3_ cells. (**A**) Representative whole-cell *I*_h_ traces obtained in the absence (a) and presence of 0.1 μM DEX (b), 0.3 μM DEX (c), or 1 μM DEX (d). The upper part shows the voltage protocol delivered. In (**B**), the activation time courses of *I*_h_ taken in control (a) and during cell exposure to 0.1 μM DEX (b), 0.3 μM DEX (c), and 1 μM DEX (d) was goodness-of-fit by single exponential (indicated by smooth lines) with a value of 683, 825, 1011, and 1406 msec, respectively. In (**C**), the time constant (τ) of the relative block by 0.1 or 0.3 μM DEX were fitted by a single exponential with a value of 163 or 143 msec (indicated by smooth lines), respectively. The relative block of *I*_h_ was evaluated by dividing the DEX-sensitive current by the current obtained in the control (i.e., (*I*_control_−*I*_DEX_)/*I*_control_). In (**D**) the reciprocal of time constant (i.e., 1/τ) of relative block versus the DEX concentration was plotted. Data points shown in open squares were least-squares fitted by a linear regression, indicating that there is a molecularity of one. From the minimum reaction scheme described in the Materials and Methods, blocking (*k*_+1_*) or unblocking (*k*_−1_) rate constants for the DEX-induced block of *I*_h_ was calculated to be 2.419 s^−1^μM^−1^ or 4.77 s^−1^, respectively. Mean ± SEM (*n* = 7–10 for each point). (**E**) Concentration-dependent inhibition of DEX on *I*_h_ in response to the hyperpolarizing voltage step (mean ± SEM; *n* = 9 for each point). Current amplitude was measured at the end of each hyperpolarizing pulse from −40 to −110 mV with a duration of 2 s. The continuous line overlaid onto the data was least-squares fitted by the Hill equation, as elaborated in the Materials and Methods.

**Figure 2 ijms-21-09110-f002:**
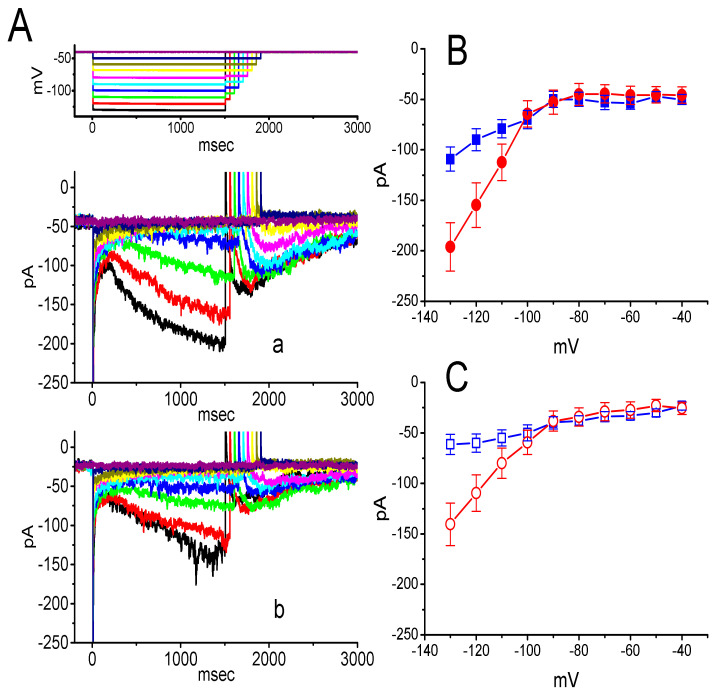
Inhibitory effects of DEX on averaged *I*–*V* relationship of *I*_h_ measured from GH_3_ cells. In these experiments, we bathed cells in Ca^2+^-free, Tyrode’s solution and filled the recording pipette by using K^+^-containing solution. (**A**) Representative whole-cell *I*_h_ traces obtained in the absence (a) and presence of 1 μM DEX (b). The uppermost part indicates the voltage protocol applied. Panels (**B**) (in the absence of DEX, closed symbols) and (**C**) (in the presence of 1 μM DEX, open symbols), respectively, show averaged *I*–*V* relationships of *I*_h_ measured at the beginning (square symbols) or end (circle symbols) of hyperpolarizing pulses. Each point indicates the mean ± SEM (*n* = 9). (**D**) Effect of DEX on the steady-state activation curve of *I*_h_. Continuous curves were optimized according to a modified Boltzmann function as detailed in the Materials and Methods. Closed symbols are controls and open symbols were obtained during the exposure to 1 μM DEX. Each point is the mean ± SEM (*n* = 7). Of note, there is a leftward shift in the activation curve of *I*_h_ during the exposure to DEX.

**Figure 3 ijms-21-09110-f003:**
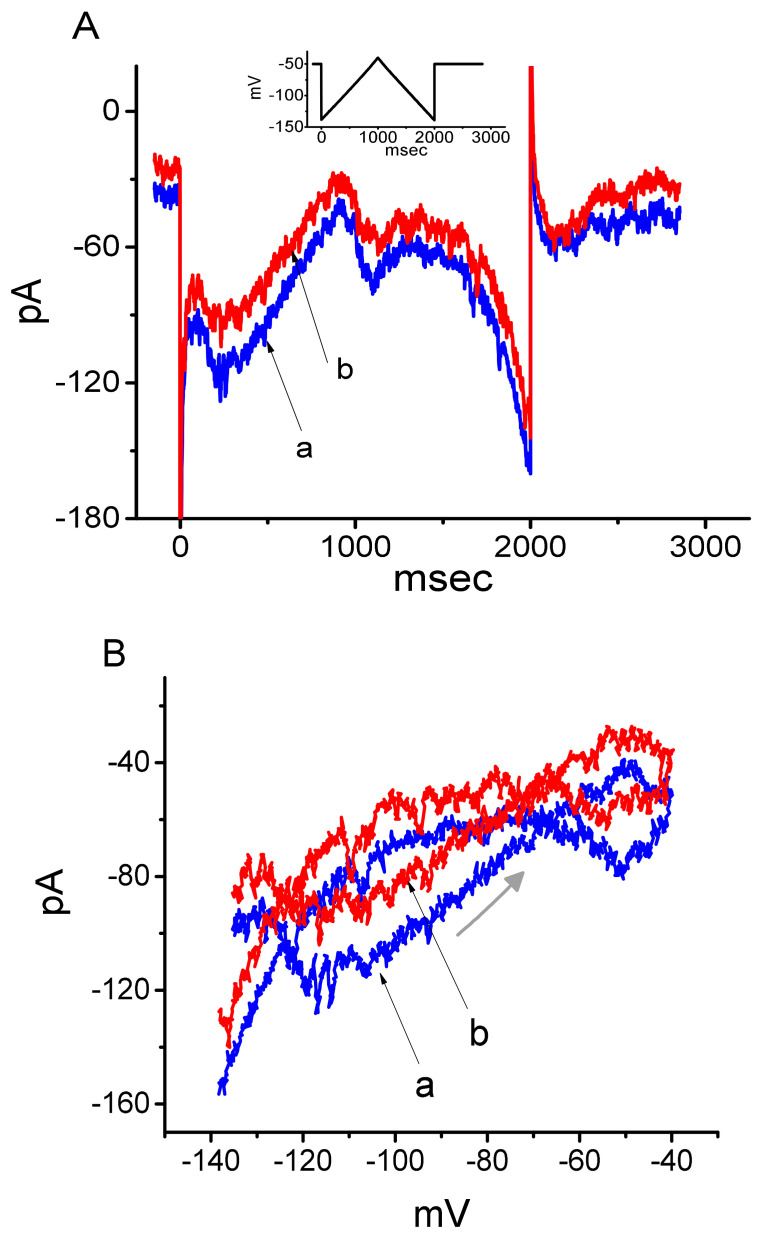
Effect of DEX on the voltage hysteresis measured from GH_3_ cells. (**A**) Representative current trace elicited by slow triangular (i.e., upsloping and downsloping) ramp command between −140 and −40 mV with a duration of 2 s. Inset in (**A**) is the voltage protocol applied. Current trace labeled a is control and that labeled b was taken in the presence of 1 μM DEX. (**B**) Voltage hysteresis (i.e., forward and reverse current versus voltage relationship) of *I*_h_ recorded from the absence (a) and presence of 1 μM DEX (b). Arrow indicates the direction of *I*_h_ trajectory in which time passes. (**C**) Summary bar graph showing the effect of DEX at a concentration of 0.3, 1, or 3 μM on the Δarea of voltage hysteresis (mean ± SEM; *n* = 9 for each bar). * Significantly different from control (*p* < 0.05).

**Figure 4 ijms-21-09110-f004:**
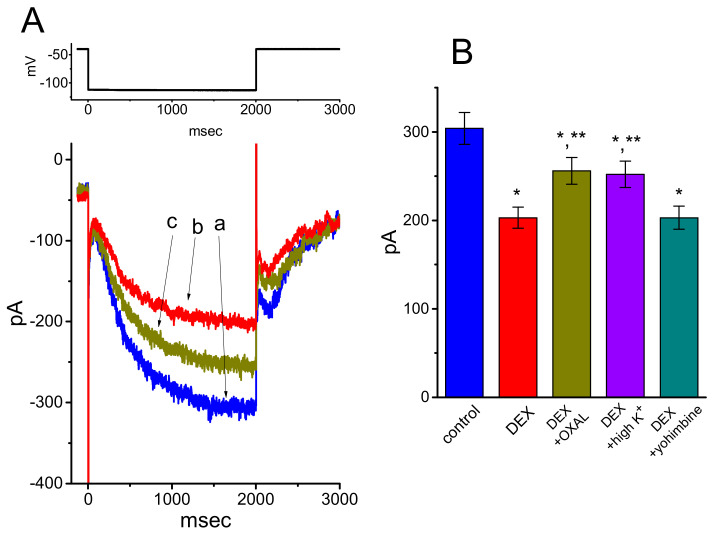
Effect of DEX, DEX plus oxaliplatin (OXAL), DEC plus high-K^+^ solution, or DEC plus yohimine on *I*_h_ amplitude in GH_3_ cells. (**A**) Representative *I*_h_ traces elicited by membrane hyperpolarization (as indicated in the upper part of (A)). a: control; b: 1 μM DEX; c 1 μM DEX plus 30 μM OXAL. (**B**) Summary bar graph showing the effects of DEX (1 μM), DEX (1 μM) plus OXAL (30 μM), DEX (1 μM) plus high-K^+^ (20 mM) solution, and DEX (1 μM) plus yohimbine (10 μM) on *I*_h_ amplitude (mean ± SEM; *n* = 8 for each bar). Current amplitude was measured at the end of 2-s hyperpolarizing step from −40 to −110 mV. * Significantly different from control (*p* < 0.05) and ** significantly different from 1 μM DEX alone group (*p* < 0.05).

**Figure 5 ijms-21-09110-f005:**
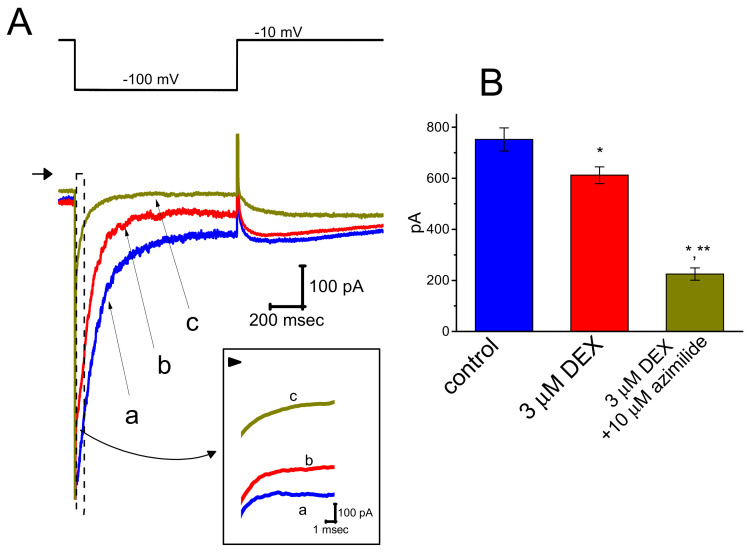
Effect of DEX on *erg*-mediated K^+^ current (*I*_K(erg)_) in GH_3_ cells. In these experiments, we immersed cells in high-K^+^, Ca^2+^-free solution and, during the measurements, we then filled the recording pipette by using the K^+^-containing solution. (**A**) Representative *I*_K(erg)_ traces obtained in the control (a), and during cell exposure to 3 μM DEX (b) or to 3 μM DEX plus 10 μM azimilide (c). The upper part is the voltage protocol used, the arrowhead is the zero current level, and calibration mark shown at the right lower corner applied all current traces. Inset indicates an expanded record shown in dashed box of (**A**), and arrowhead is zero current level. (**B**) Summary bar graph showing the inhibitory effects of DEX and DEX plus azimilide on the amplitude of *I*_K(erg)_ (mean ± SEM; *n* = 8 for each bar). Current amplitude (i.e., deactivating *I*_K(erg)_) was measured at the beginning of 1-s hyperpolarizing pulse from −10 to −100 mV. * Significantly different from control (*p* < 0.05) and ** significantly different from 3 μM DEX alone group (*p* < 0.05).

**Figure 6 ijms-21-09110-f006:**
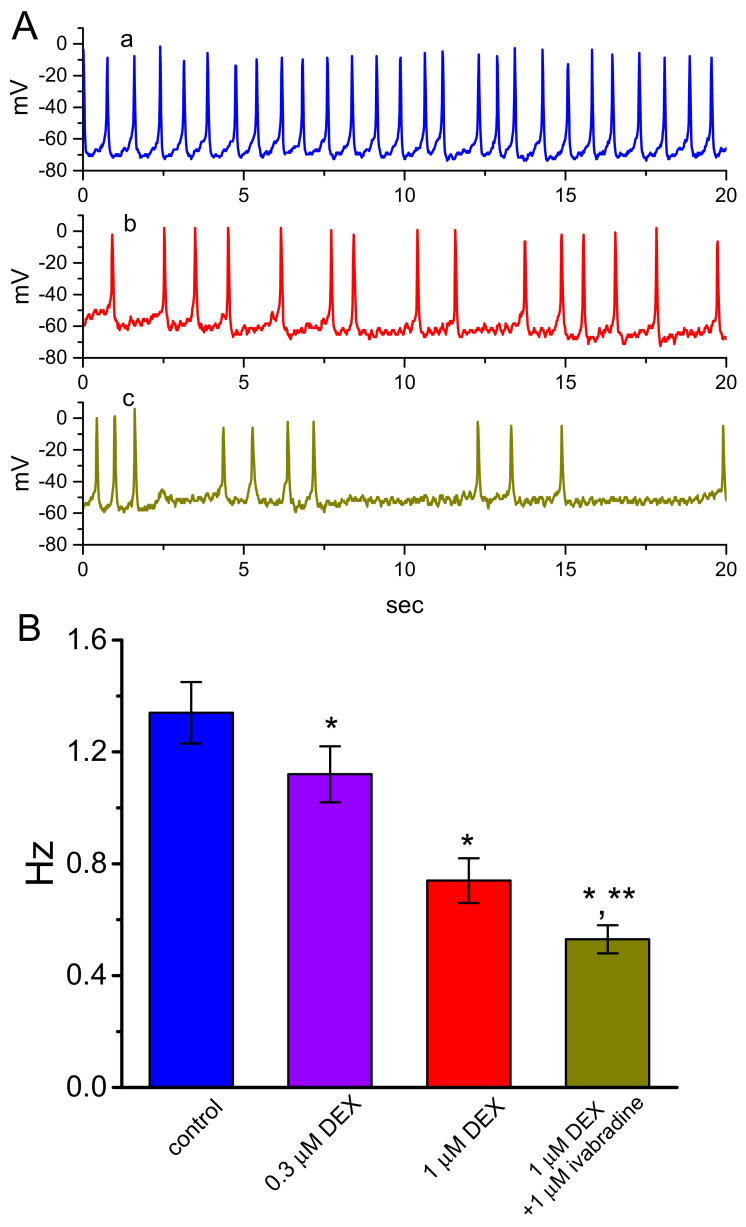
Effect of DEX and ivabradine on spontaneous action potentials (APs) in GH_3_ cells. Cells were bathed in normal Tyrode’s solution containing 1.8 mM CaCl_2_ and the pipette was filled with K^+^-containing solution. Once whole-cell configuration was achieved, we rapidly switched to the mode used for current-clamp voltage recordings. (**A**) Representative potential traces obtained in the control (a), and during the exposure to 1 μM DEX (b) or to 1 μM DEX plus 1 μM ivabradine (c). (**B**) Summary bar graph showing the effects of DEX and DEX plus ivabradine on the firing frequency of spontaneous APs seen in GH_3_ cells (mean ± SEM; *n* = 11 for each bar). * Significantly different from control (*p* < 0.05) and ** significantly different from 1 μM DEX alone group (*p* < 0.05).

**Figure 7 ijms-21-09110-f007:**
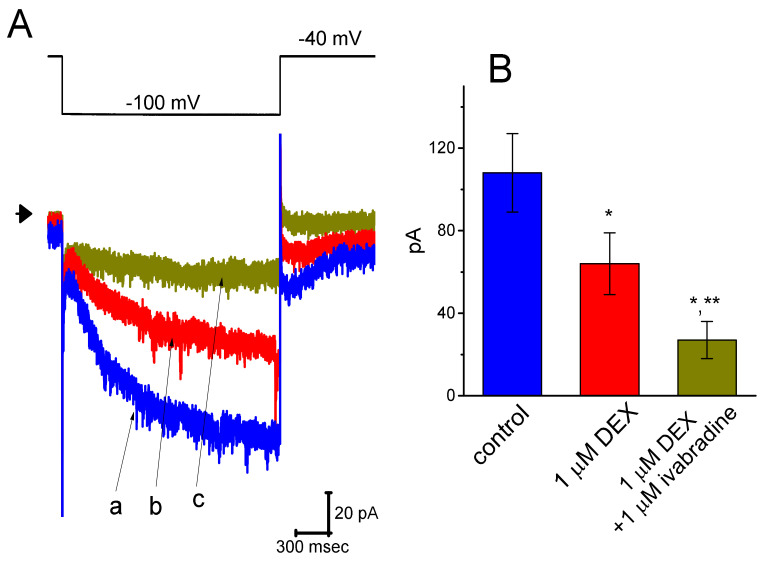
Inhibition by DEX and ivabradine of hyperpolarization-elicited *I*_h_ in pheochromocytoma PC12 cells. Cells were bathed in Ca^2+^-free, Tyrode’s solution and the pipette was filled with K^+^-containing solution. (**A**) Representative *I*_h_ trace obtained in the control (a) and during the exposure to 1 μM DEX alone (b) and 1 μM DEX plus 1 μM ivabradine (c). The upper part denotes the voltage protocol used, and arrowhead in the left side of traces is the zero current level. (**B**) Summary bar graph showing the effects of DEX and DEX plus ivabradine on *I*_h_ amplitude present in PC12 cells (mean ± SEM; *n* = 7 for each bar). Current amplitude was obtained at the end of each hyperpolarizing pulse from −40 to −100 mV. * Significantly different from control (*p* < 0.05) and ** significantly different from 1 μM DEX alone group (*p* < 0.05).

**Figure 8 ijms-21-09110-f008:**
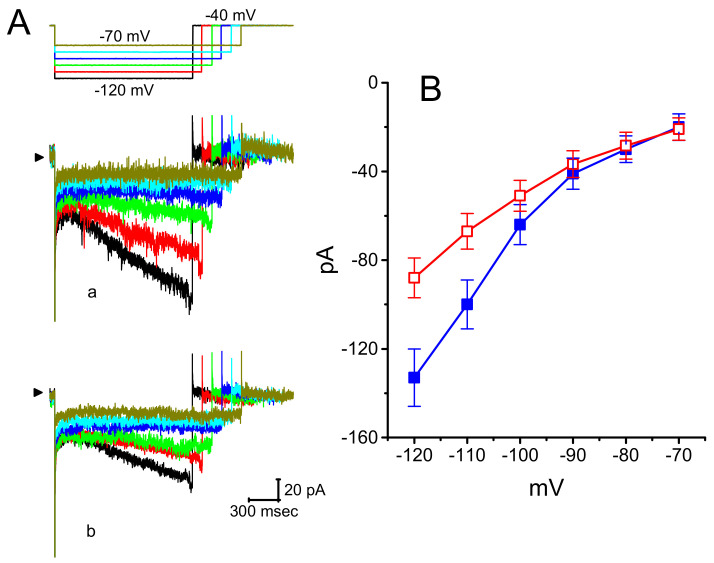
Effect of DEX on *I*–*V* relationship of *I*_h_ recorded from pheochromcytoma PC12 cells. (**A**) Representative *I*_h_ traces elicited by different hyperpolarizing steps from a holding potential of −40 mV (as indicated in the uppermost part of (**A**)). a: control; b: in the presence of 1 μM DEX. (**B**) Averaged *I*–*V* relationships of *I*_h_ amplitude taken from the absence (■) or presence (□) of 1 μM DEX (mean ± SEM; *n* = 8 for each point). Current amplitude was measured at the end of each hyperpolarizing step potential. Note that, in PC12 cells, DEX was capable of suppressing *I*_h_ measured at the voltages ranging between −100 and −120 mV.

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
