# Peer review of "Effectiveness in Block by Dexmedetomidine of Hyperpolarization-Activated Cation Current, Independent of Its Agonistic Effect on α_2_-Adrenergic Receptors"

_ijms, 2020, doi:10.3390/ijms21239110_

Round 1

Reviewer 1 Report

The authors are presenting the effect of dexmedetomidine on blocking the cation current in a well organized manner. Therefore I would like to recommend the acceptance of this article.

Author Response

Thanks for the comments by the reviewer.

Reviewer 2 Report

In this manuscript, the authors describe inhibitory effects of dexmedetomidine (DEX), an alpha2 adrenoceptor agonist, on hyperpolarization-activated cation currents (Ih). To use culture cells with Ih, the authors demonstrated that DEX inhibited Ih in a concentration-dependent manner. Overall, the electrophysiological experiments are well performed and the results are clear. Following points should be addressed to improve the manuscript.

  1. Methods (l183-184); Which data did the authors apply Kruskal-Wallis test to? Please specify the use in Methods or the figure legend.

  1. Results (l200-201); Application of 1uM DEX inhibited Ih from 193.8 pA to 188.7 pA (n=11). However, the inhibition was much larger in Fig1A. Is FIg.1A an untypical example?

  1. Fig.2D; The label 'D' is missing and the position of Fig.2D is unusual.

  1. Fig.4; Does OXAL or high K+ itself without DEX have any effects on Ih?

  1. Fig.5; The analysis of erg-K+ current amplitude is not clear. The authors measured the erg-current size at the beginning of 1 sec hyperpolarizing pulse from -10 to -100 mV (l341-342, the figure legend). However, the capacitance in Fig.5A masked the appearance of the inhibition, in particular in the presence of azimilide. Please add a time-expanding trace to Fig.5.

Author Response

  1. Methods (l183-184); Which data did the authors apply Kruskal-Wallis test to? Please specify the use in Methods or the figure legend.

 Ans: Thanks for the reviewer’s comment. In fact, in the present experiments, we did not apply the data for non-parametric Kruskal-Wallis test. Hence, we decided to remove the sentence from the revised manuscript.

2. Results (l200-201); Application of 1uM DEX inhibited Ih from 193.8 pA to 188.7 pA (n=11). However, the inhibition was much larger in Fig1A. Is FIg.1A an untypical example?

 Ans: Thanks for bringing our attention. We made mistakes. The data in the text of the revised manuscript were corrected and appropriately changed. That is, “For example, the addition of 1 mM DEX decreased Ih amplitude from 253.3±12.5 to 138.7±9.8 pA (n=11, P<0.05). After washout of the agent, current amplitude returned to 249.6±11.2 pA (n=9, P<0.05).“ (page 11, lines 230-232).

3. Fig.2D; The label 'D' is missing and the position of Fig.2D is unusual.

 Ans: Figure 2D was redrawn for clarity. The label “D” was included as well.

4. Fig.4; Does OXAL or high K+ itself without DEX have any effects on Ih?

Ans: OXAL was recently reported to elevate Ih amplitude (Resta et al., 2018), challenging cells with high K+ solution has been shown to increase Ih amplitude (McCormick and Pape, 1990; Hu et al., 2016; Chang et al., 2020),

5. Fig.5; The analysis of erg-K+ current amplitude is not clear. The authors measured the erg-current size at the beginning of 1 sec hyperpolarizing pulse from -10 to -100 mV (l341-342, the figure legend). However, the capacitance in Fig.5A masked the appearance of the inhibition, in particular in the presence of azimilide. Please add a time-expanding trace to Fig.5.

Ans: As advised by the reviewer, Figure 5A was redone and inset was correspondingly included in the Figure 5. The legend in Figure 5A was hence included in the revised manuscript (lines 643-644)